# A Randomized Controlled Study to Test Front-of-Pack (FOP) Nutrition Labels in the Kingdom of Saudi Arabia

**DOI:** 10.3390/nu15132904

**Published:** 2023-06-27

**Authors:** Soye Shin, Ada Mohammad Alqunaibet, Reem F. Alsukait, Amaal Alruwaily, Rasha Abdulrahman Alfawaz, Abdullah Algwizani, Christopher H. Herbst, Meera Shekar, Eric A. Finkelstein

**Affiliations:** 1Program in Health Services and Systems Research, Duke-NUS Medical School, Singapore 169857, Singapore; eric.finkelstein@duke-nus.edu.sg; 2Public Health Authority, Riyadh 11176, Saudi Arabia; alqunaibetm@moh.gov.sa (A.M.A.); arruwaily@cdc.gov.sa (A.A.); rafawaz@cdc.gov.sa (R.A.A.); argwizani@cdc.gov.sa (A.A.); 3Department of Community Health Sciences, King Saud University, Riyadh 11362, Saudi Arabia; ralsukait@ksu.edu.sa; 4Health, Nutrition and Population Global Practice, World Bank Group, Washington, DC 20433, USA; cherbst@worldbank.org (C.H.H.); mshekar@worldbank.org (M.S.)

**Keywords:** front-of-pack labeling, Chilean warning labels, Nutri-Score, diet quality, online grocery shopping

## Abstract

One common strategy for governments to tackle the non-communicable disease (NCD) epidemic is front-of-package (FOP) nutrition labeling. The Kingdom of Saudi Arabia (KSA) is considering implementing a new FOP label that is based on either France’s Nutri-Score (NS), which labels all foods (A = healthiest to E = least healthy) based on overall nutritional quality, or the Chilean warning label (WL) approach, which identifies foods to avoid based on select nutritional characteristics. Using a fully functional online grocery store, this study aimed to test these two promising FOP strategies by randomizing 656 KSA adults into one of the three versions of the store to complete a hypothetical grocery shop: no-label (control), NS, and WL. The NS was modified with a sugar percentage tag given that reducing sugar consumption is one of KSA’s public health goals. We found that both modified NS labels and Chilean warning labels positively influenced food and beverage choices among KSA participants, but there were differential effects across the two labels. Relative to the control, NS improved the overall diet quality of the shopping baskets, measured by the weighted (by the number of servings) average NS point (ranging from 0, least healthy, to 55, healthiest), by 2.5 points [95% CI: 1.7, 3.4; *p* < 0.001], whereas results for WL were not statistically significant (0.6 points [95% CI: −0.2,1.5]). With respect to each nutritional attribute, we found that NS reduced sugar intake per serving, whereas WL was effective at decreasing energy and saturated fat intake per serving from food and beverages purchased. Our results suggest that the NS approach that identifies the healthiness of all foods using a holistic approach appears preferable if the purpose of the label is to improve overall diet quality as opposed to addressing select nutrients to avoid.

## 1. Introduction

The health and economic burden of non-communicable diseases (NCDs) globally is large and growing [1,2,3]. According to the World Health Organization, the annual number of deaths due to NCDs is 41 million, which is equivalent to 71% of global deaths [2]. Focusing on the well-established relationship between NCDs (e.g., diabetes, cardiovascular disease, cancer, etc.) and diet [4,5,6,7,8], many governments have attempted to tackle the NCD epidemic by implementing interventions aimed at promoting healthier food choices. One increasingly common strategy is front-of-package (FOP) nutrition labeling [9,10,11,12,13,14,15]. FOP labels are intended to provide consumers with salient and easy-to-comprehend information on the nutritional quality of foods, complementing the existing nutrition facts panel (NFP) on the back of the products [16,17,18].

FOP labels can generally be classified as reductive or interpretive. Reductive labels present a subset of relevant information without interpretation, such as calories per serving and calories per day for a healthy diet. Interpretive labels use nutritional information to convey a message to consumers as to the underlying healthiness of the product in the dimensions considered. Interpretive FOP labels can focus solely on identifying foods that are healthier (i.e., positive labels), such as in Singapore’s healthier choice symbol (HCS) labels [19]; on foods that are less healthy (i.e., negative labels), such as Chile’s warning labels [14]; or both (i.e., graded labels), such as France’s Nutri-Score label [20], which grades all foods from A to E on overall nutritional quality. Which label is most effective at influencing overall diet quality for a given population is ultimately an empirical question. Further, each label is likely to have both intended and unintended consequences. For example, because a positive label offers a signal that a product is “healthier” [21], it could induce consumers to overconsume labeled products, thus resulting in an improvement in diet quality and, ironically, an increase in total calories [22]. For this reason, countries have moved away from solely showing positive FOP labels in favor of graded approaches or warning labels. Graded labels, which condense the complicated information on the NFP into a single summary score, are likely to be more effective at improving overall diet quality [23,24] than warning labels. This results because graded labels identify the healthiness of all foods. By contrast, warning labels only identify the worst foods to avoid and are often based only on a single nutrient, such as sugar or sodium. Not only do they not help consumers determine the overall healthfulness of a given food, but they also do not identify which of the non-targeted foods are healthier.

Despite the limitations, both graded and warning FOP labels have been shown to be effective relative to a no-label control condition [25,26,27]. However, few head-to-head studies exist directly comparing the two labels, and those that do are limited to select food categories [28]. For instance, Egnell et al. (2018) [29] assessed the relative effectiveness of five FOP labels in 12 countries. They found that Nutri-Score performed best in helping participants correctly rank products according to overall nutritional quality. However, their analyses were based on simple choice experiments with only three food categories (pizzas, cakes, and breakfast cereals). Therefore, results may not generalize to other products within these countries or to other populations where individuals may have a different knowledge base and/or different diet and health preferences.

As with many countries, Saudi Arabia (KSA) has observed a dramatic rise in NCDs. Comparing data between 2008 and 2017, the prevalence of diabetes, cardiovascular disease, and cancer increased to 51%, 47%, and 96%, respectively [30], mainly due to poor diets, physical inactivity, and high smoking rates, and these increases likely continued during the COVID-19 pandemic [31]. KSA is now considering an FOP label to complement other policies, such as beverage taxes [32,33], aimed at curbing the rise in NCDs. The purpose of this study was to test two promising FOP strategies. One is a slight variant of the Nutri-Score label used in multiple countries in Europe. Since reducing sugar consumption is one of KSA’s public health goals [34,35], the Nutri-Score (NS) label was modified to also show the percentage of sugar per serving on the right side of the label (Figure 1, top panel), thus acting similarly to the warning label for this nutrient. The design of the label looks similar to Singapore’s new FOP label for beverages, called Nutri-Grade, which is shown to positively influence beverage choices [36]. The second is the Arabic-language version of Chile’s stop-sign warning labels (WL) for products that are high in calories, sugar, sodium, and saturated fat (English versions are shown in Figure 1, bottom panel).

We chose a holistic measure of nutritional quality, the weighted average Nutri-Score point, as the primary outcome. Although not free from concerns [37], it is a reasonable summary indicator of the healthiness of the shopping basket, is publicly available, and has been shown to be correlated with key measures of diet quality [38,39] and NCD-related health outcomes [40,41,42]. However, to explore the effect on specific nutrients, we complemented the primary outcome with secondary outcomes that include calories and the nutrients that make up the inputs into the NS algorithm and the target nutrients of WL as well.

We expect that both labels would positively influence food-purchasing patterns relative to a no-label control arm. We hypothesized that the modified Nutri-Score label (NS) would perform best when it comes to overall diet quality, given that is its primary focus. Warning labels (WL) are expected to be more effective at reducing the nutrients (and calories) that are the target of the labels, including sugar intake. This results because the stop sign warning is a stronger signal than simply noting the sugar percentage per serving, as it appears on the NS label. We tested these hypotheses with KSA shoppers using a randomized controlled trial design and a fully functional web grocery store where shoppers completed a one-time hypothetical grocery shop. We investigated the effectiveness of the labels not only on foods and beverages but also on beverages alone, given that sugar-sweetened beverages are a primary contributor to obesity and NCDs [43,44]. The results of this study will help policymakers determine which strategy is likely to be most effective in the KSA.

## 2. Materials and Methods

### 2.1. Online Grocery Store

The study took advantage of an experimental online grocery store developed for research purposes (https://nusmartbulletin.wordpress.com/ (accessed on 26 May 2023)) [45]. This online store was designed to mirror a commercial online grocery store but to be highly flexible in testing various tools and interventions aimed at improving diet quality. Participants could add and remove products from their online grocery cart and review their cumulative total cart cost. They could also sort products by name and price, with the default showing products from the least expensive to the most expensive.

For this study, products were primarily selected from an online store of a large Saudi supermarket, called Danube (https://danube.sa/ (accessed on 26 May 2023)) [46], to represent as many food categories as possible and subsequently reviewed by the KSA Public Health Authority. We then collected product-specific information from various online sources, including the Danube website. We dropped the products for which we could not find nutritional information. As a result, the store contained 1969 food and beverage products commonly purchased in Saudi Arabian supermarkets. Food items were classified into one of 23 categories (on average, 199 unique products per category), and then by subcategories (on average, 69 unique products per subcategory) within the broader category (e.g., dairy and eggs were subcategorized as butter and margarine spreads; cheese slices, blocks, and cubes; cheese spreads and labneh; cream, creamer, and condensed milk; fresh cream, etc.). The list of the category-subcategory pairs used for the store is found in Appendix A. All products include the item’s name in the local language, a picture of the item, retail price, product description, and nutritional information available via click-through. Figure 2 presents a screenshot of the default version of the grocery store webpage, which was used for the control arm of the study.

### 2.2. Experimental Design

The study was a three-arm randomized controlled trial (RCT) design with three versions of the grocery store (the no-label control, the NS, and the WL arms; see Figure 3). The five Nutri-Score grades (A, healthiest, to E, least healthy) were determined based on NS points ranging from −15 (healthiest) to 40 (least healthy) that are calculated according to the Nutri-Score point system. Relying on the British Food Standard Agency Nutrient Profiling System [13,20,47], the Nutri-Score point system used in this effort (prior to a revision in September 2022) assigned points to each product based on levels of seven nutritional components per 100 g or 100 mL. Greater amounts of the four components that are typically overconsumed reduce the total score. These are energy, sugar, sodium, and saturated fat. Greater amounts of fruits and vegetables, protein, and dietary fiber, all of which are correlated with good health, increase the score. To compute the sugar percentage, we used a food-subcategory average serving size, considering that the amount per serving (i.e., the amount of a food/drink for one sitting) determined by each manufacturer can be arbitrary. Unlike many prior studies [48,49], we displayed the modified NS on both packaged products and unpackaged fresh fruits and vegetables. Assigning the label to both pre-packed and fresh foods removes a potential unintended consequence of FOP labels that could occur if shoppers opt for labeled processed products in lieu of unlabeled fresh products [50]. The WL showed an intuitive black stop sign with the simple message “High in [Nutritional attribute X]” in the center. Each product was assigned WL for each target nutritional attribute when the corresponding energy/nutrition content per 100g or 100 mL is over the Chilean nutrition thresholds [14].

In Figure 3, Arm 2, (NS) store, all 1969 products were labeled with NS. The percentages of products with each NS grade were 32% (A), 13% (B), 18% (C), 18% (D), and 18% (E). In Figure 3, Arm 3, (WL) store, 65% of products received at least one WL (45% calories, 26% sugar, 27% sodium, and 29% saturated fat). Among 250 beverages in the NS store, the percentages were 31% (B), 13% (C), 8% (D), 48% (E), and no NS A beverage as our store did not contain water. In the WL store, 38% of beverages had at least one WL (25% had calories, 36% sugar, 13% sodium, and 8% had saturated fat).

### 2.3. Participants and Procedures

Participants were recruited from July to August 2021 from the Kantar online web panel of Saudi Arabian citizens and residents. Prospective participants were asked to complete an online screener to determine their eligibility. They were eligible if they were aged 18 years or older, were KSA residents, able to read and write in Arabic, and were the primary grocery shopper for their household. Those interested and eligible were asked to complete an online consent form. Participants were made aware that there were multiple versions of the online grocery store but were not told of the nature of the study; those who consented were exposed only to the version of the store into which they were randomized.

Once they consented, participants were directed to a baseline survey collecting their demographic characteristics, including age, sex, education level, housing type, monthly household income, and the underlying diet-related health conditions of their household members (e.g., overweight, obesity, diabetes, and hypertension). Upon completion of the baseline survey, participants were randomly assigned to one of the three study arms with equal probability and landed in a randomly chosen food subcategory to start shopping. Participants were then asked to complete a single hypothetical shopping trip as though all meals and snacks for their household members for the next 7 days would be made from foods and beverages purchased from this shopping trip. In efforts to encourage shoppers to take the task seriously, they were also required to spend a minimum of 150 Saudi riyals (SAR, equivalent to 40 US dollars) per household member and to shop in at least four food categories. This minimum amount was determined based on the average per capita monthly spending on foods and beverages (excluding consumption away from home) reported in the 2018 KSA Household Expenditure Survey, with inflation and taxes accounted for. Those who were assigned to the NS arm were shown a brief introductory video about the modified NS label. They were able to watch this video again at any time during their shopping trip. We did not show a similar video to participants assigned to the WL arm because we believed that messages on WL (‘high in [nutritional component X]’) are relatively straightforward. Panelists who completed the study received compensation in points according to Kantar’s in-house incentive protocols.

The use of a web panel enabled us to collect data relatively quickly, at low cost, and from a geographically diverse population of shoppers in KSA. Once participants completed their shopping and clicked on “check-out,” they were asked to take a brief post-study survey that included questions that tested how well they understood the labels and video (if applicable), as well as an open-ended feedback question.

A power calculation revealed that 602 participants were required to detect a standardized effect size of 0.3 in outcomes between arms, assuming a two-tailed *t*-test, power of 0.8, alpha of 0.05, and 10% attrition. Recruitment continued until this threshold was met. This study was exempted from a full review both by the National University of Singapore Institutional Review Board (IRB) Reference Code: NUS-IRB-2020-794 and the KSA Public Health Authority IRB Reference Code: SCDC-IRB-A034-2021 because we did not collect personally identifiable information. The procedures followed were in accordance with the ethical standards of the responsible institutional or regional committee on human experimentation or in accordance with the Helsinki Declaration of 1975 as revised in 1983. The trial was registered on the Clinical Trial Registry under the ID NCT05007184, on 16 August 2021.

### 2.4. Statistical Analyses

#### 2.4.1. Outcome Variables

The primary outcome was a holistic measure of diet quality: the average NS points of the shopping basket, weighted by the number of servings. Because the NS points range from −15 to 40, to ease the interpretation of the results, we reversed and shifted the NS point such that it lies in between 0 and 55, with 0 being the least healthy score and 55 the healthiest. For instance, if a participant’s grocery basket contains beverage A (1 serving, NS point 13) and food B (3 servings, NS point 30), then we compute the weighted average of NS points by using the following formula: $\frac{\sum \left(NS_{i}\times servings_{i}\right)}{Total\;number\;of\;servings}=\frac{\left(13\times 1\right)+(30\times 3)}{4}=25.75$.

The secondary outcome measures of diet quality focused on (1) total and (2) weighted (by the number of servings) average calories, sugar, sodium, and saturated fat per serving of the grocery baskets. The per-serving measures were calculated in the same way as for the primary outcome variable. The reason we are interested in both total and per-serving changes is to examine whether the labels induce people to purchase fewer nutrients or energy per serving but more total nutrients or energy, which has been shown to occur for some labels [24]. Per-serving estimates were generated by dividing the totals by an estimated number of servings based on the average serving size of all foods in the subcategory.

#### 2.4.2. Estimation

To test the hypotheses, we employed ordinary least squares (OLS) regressions with robust standard errors. The regression specification is as follows:$$Y_{i}=\beta_{0}+\beta_{1}NS_{i}+\beta_{2}WL_{i}+X^{\prime }\gamma +\varepsilon_{i}$$
where $Y_{i}$ is an outcome variable of interest observed for participant i, the constant term $\beta_{0}$ represents the mean outcome value in the control arm (no-label). The coefficients $\beta_{1}$ and $\beta_{2}$ represent the incremental effect on the outcome due to the NS ($NS_{i}$) and WL ($WL_{i}$) arms, respectively. The difference between $\beta_{1}$ and $\beta_{2}$ estimates the incremental/decremental effect of the WL relative to the NS. X is a vector of covariates that includes age, dummies for females, household size, high education level (university degree and above), high income (monthly household income of SAR 15,000 and above), participants’ most important grocery shopping considerations (health, price, taste, variety, and convenience (omitted category)), and a dummy for no diet-related health condition (e.g., obesity, diabetes) among household members. $\varepsilon_{i}$ is the error term. We also ran this model, limiting the data to beverages, recognizing that KSA has focused its tax strategy on beverages and could consider a similar approach to the FOP labels on beverages.

To gain further insight into what is driving differential effectiveness, we quantified the following across the arms: (1) the proportion of products in a grocery basket that are subject to the warning labels (and labeled in the WL arm); (2) the weighted average NS points of the products that are subject to the warning labels; and (3) the weighted average NS points of the products that are *not* subject to the warning labels (i.e., non-warning-labeled products). We hypothesized that the proportion of warning-labeled products purchased is smallest in the WL arm, followed by NS arm, followed by the control. Contrarily, we hypothesized that the NS points of the warning labeled and non-warning-labeled products are highest in the NS arm, followed by the WL arm, followed by control.

## 3. Results

### 3.1. Participants

Figure 4 outlines participant flow and randomization. In total, 4132 individuals filled out the screener; 1421 participants were eligible, consented to participate, and were randomized into one of the three arms; and 656 completed all study components and were thus included in the final analysis sample. We compared demographic characteristics between the ‘study completion group’ (n = 656) and the ‘dropout groups’ (n = 756). The results are reported in Appendix A. The two groups had statistically similar distributions of age, sex, and household size, but the latter group showed slightly smaller proportions of those who have a high monthly household income, a university degree or above, and no household member with underlying health conditions (statistically significant at *p* < 0.001).

Table 1 presents descriptive statistics of participants by study arm and total. Across the three arms, the age of participants ranged from 18 to 56 years, with a mean of 32. Roughly half of the participants were female. Participants were generally highly educated, with 81% achieving the educational attainment of “university degree or above”, 44% having a monthly household income of “15,000 SAR and above”, and 59% reporting having at least one household member with underlying health conditions. In terms of drivers of shopping behavior, taste was most important (38%), followed by health (24%), price (17%), variety (13%), and convenience (8%). Compared with national statistics, our sample consists of higher proportions of female, young (age 25 to 44), and wealthier participants, which is not surprising given that women tend to do the grocery shopping for their households in KSA, and younger and wealthier individuals have higher access to online shops [51,52]. The characteristics of the participants were statistically similar across the arms, except that there were fewer highly educated participants in the NS arm and fewer female participants and smaller household size in the WL arm, relative to the control arm. Including or excluding these three variables made no meaningful change to the estimated treatment effects.

### 3.2. The Effects of the FOP Labels on Diet Quality for All Food and Beverages

Looking into the composition of the shopping baskets, we found that “fresh vegetables” (11.3%), “legumes, nuts, and seeds” (10%), and “rice” (8%) were the three subcategories that had the highest spending for each basket, in order of frequency. This composition generally holds across arms except that “rice” was the most frequently occurring subcategory (10.8%) in the no-label arm and “cold cereals” ranked third (6.6%) in the NS arm. Before looking into the regression results, we report the descriptive statistics of each outcome variable by arm in Table 2.

Table 3 and Table 4 summarize the effects of the FOP labels on each outcome variable for food and beverage products purchased (1) relative to the control arm and (2) between the two FOP labels. The full regression results are found in Appendix A. Consistent with our hypothesis, both the NS and the WL increased the weighted average NS points relative to the no-label control arm (2.5 points [95% CI: 1.7, 3.4] and 0.6 points [95% CI: −0.2, 1.5], respectively), but the difference was statistically significant only for the NS arm at *p* < 0.001. This effect of the NS label equates to a roughly one letter grade improvement in diet quality. Among the covariates, those who are older, consider “health or variety” as the most important factor for grocery shopping (relative to “convenience”), and have no household members with underlying health conditions were associated with higher diet quality in the grocery baskets.

In terms of per-serving energy and nutrients, NS reduced sugar intake per serving, whereas WL reduced calories and saturated fat per serving relative to the control arm. In the NS arm, which included the tag with the percentage of sugar per serving, shoppers purchased less sugar (g) per serving, on average, by 2.1 g [95% CI: −2.9, −1.4; *p* < 0.001]. The amount of the reduction in the NS arm is significantly greater than the (statistically insignificant) sugar reduction in the WL arm (a difference of 1.6 g [95% CI: −2.4, −0.9]; *p* < 0.001). The reduction in saturated fat in the WL arm was greater than the reduction in the NS arm (difference of 30.7 g [95% CI: −3.5, 64.9] but not statistically significant with *p* = 0.078), whereas the effect on calories for the WL arm was not statistically different from that of the NS arm (difference of 68.7 kcal [95% CI: −33.5, 170.8]). None of the labels reduced sodium (mg) per serving. The results for total calories and the other included nutrients (Table 4) are qualitatively analogous to the per-serving calories and nutrient results in Table 3, except that the NS reduced the total sodium (mg) intake (only statistically significant at *p* = 0.059) of the shopping basket.

The mean percentage of the warning-labeled products purchased was 71% in the control arm, 62% in the NS arm, and 68% in the WL arm. All differences are statistically significant (*p* < 0.001). We also found that the weighted average NS point (ranging from 0 to 55) among products that are *not* subject to receiving warning labels was the highest in the NS arm (36.6), followed by the WL arm (34.4), and the no-label control (33.5). The same results hold when we compare the weighted average NS points among the products that are subject to warning labels. Relative to the no-label control (32.3), the NS point score was 1.8 points higher in the NS arm and 0.1 points higher in the WL arm.

### 3.3. The Effects of FOP Labels on Diet Quality for Beverages Only

We present the results on the primary outcome along with the per-serving outcomes in Table 5 and the total energy and nutrient outcomes in Table 6 for beverages only. We report the full regression results in Appendix A. On average, 11% of the total shopping basket contained beverages. Both labels are effective at improving the nutritional quality of beverages purchased relative to the no-label control. Compared with the control arm, the average NS point was higher by 2.5 points [95% CI: 1.3, 3.7; *p* < 0.001] in the NS arm and by 1.6 points [95% CI: 0.5, 2.8; *p* = 0.006] in the WL arm. The 0.9-point difference between the two labels is not statistically significant (*p* = 0.145). These two labels were also effective at reducing calories and sugar. The amount of the calories and sugar per serving reduction is 24.5 kcal [95% CI: −36.7, −12.3; *p* < 0.001] and 6.3 g [95% CI: −10.1, −2.5; *p* = 0.001] in the NS arm, and 17.9 kcal [95% CI: −29.8, −6.0; *p* = 0.003] and 5.0 g [95% CI: −8.4, −1.7; *p* = 0.003] in the WL arm, respectively. Similar to the results for food and beverage, we found that only WL decreased saturated fat intakes per serving from beverages purchased (0.2 g [95% CI: −0.5, 0.0]; only significant at *p* = 0.068) relative to the control arm. Lastly, the directions of the effects on total energy and nutrients (Table 6) were generally consistent with the results in Table 4. For all outcomes, differences between labels were not statistically significant.

The proportion of warning-labeled beverages in the baskets is 9.2% in the NS arm and 8.7% in the WL arm. Relative to the control arm (9.7%), the difference is only significant for the WL arm (*p* = 0.003), and there was some evidence of a difference between the two label arms (*p* = 0.088). In terms of the weighted average NS points among warning-labeled and non-warning-labeled products, the NS performs the best (34.2), followed by the WL (31.6), followed by the control (30.7), echoing the finding from food and beverage.

## 4. Discussion

Consistent with prior evidence, we found that both the modified Nutri-Score (NS) and the Chilean warning labels (WL) promoted healthier food and beverage choices relative to the no-label control arm. However, the effectiveness of the two labels varied depending on which outcome we focused on. Consistent with our hypothesis, the NS improved the overall diet quality of shopping baskets as measured by the weighted (by the number of servings) average NS point, whereas the WL did not. However, again, as hypothesized, the WL is generally more effective at reducing targeted nutritional attributes. Compared with the no-label control, these labels were more effective at decreasing per-serving energy (kcal) and saturated fat (g) intake. However, contrary to expectations, the reduction in sugar intake per serving was greater with the NS. This finding may be due to our modification of the NS to explicitly show the sugar percentage per serving, thus minimizing the difference between the two labels for this nutrient.

Although cross-study comparisons may be confounded by differences in participant and store characteristics, the effect size of the two labels in our study appears reasonable. A study testing the standard NS label with 290 products with 691 French subjects showed the NS score of a grocery basket (normalized by 100 kcal) in the NS arm was better by 2.65 points relative to the no-label arm [53]. This is very similar to our estimate of 2.5 points. A study evaluating WL with a choice experiment from five food groups in Mexico reported that the labels reduced energy and macronutrient consumption relative to the guideline daily amounts (GDA) label [54]. They showed mean energy and saturated fat reductions relative to the GDA label of 9.9 kcal and 0.2 g per 100g/mL, respectively. This is smaller than our estimates of 98.2 kcal and 24.6 g per serving reduction (the mean serving size is 66 g in our data) in the WL arm. This difference is not unexpected given that, unlike in their study, our control arm offers no FOP information.

We further show that both labels improved overall diet quality and reduced per-serving intakes of sugar and calories from beverages purchased. We also found no evidence that either label induced shoppers to purchase more *total* nutrients and energy. Since sugar is the main energy source for beverage products, it is likely that the labels’ positive effects on sugar reductions translate into the demonstrated effects on calories and overall diet quality. Given our design, we cannot tell whether the beneficial effects of our NS are the result of the letter grades and/or the result of explicitly tagging the sugar percentage per serving.

The heterogeneous effect of the two labels on different outcomes is an important finding. The choice of one label over another should depend on the public health goal. Our results suggest that, if the goal is to improve overall diet quality while targeting one factor to emphasize (sugar in our case), then a modified NS label as applied here should be considered. If the goal is to reduce intake of a small number of negative factors, then warning labels targeting these factors may be the optimal approach, in which case focusing on those nutrients as the primary outcomes would be appropriate. However, if there are too many nutrients targeted, efforts to avoid one nutrient could inadvertently increase the consumption of another. Our study also showed that not only did the modified NS label improve overall diet quality, but that, compared to shoppers in the WL arm, shoppers in the NS arm purchased healthier non-WL products (i.e., products that did not have a warning label in the WL arm had higher average NS points in a grocery cart) and bought a smaller percentage of WL products (products that had a warning label in the WL arm). This is consistent with the NS label providing additional information that is not available to those in the warning label arm.

None of the labels statistically reduced sodium (mg) per serving, which contrasts with our hypothesis about the effectiveness of the high-in-sodium warning label. Although few papers reported nutrient-specific effects of the Chilean warning-type labels, one study with Mexican adults found that the labels led to a reduction in sodium purchases per 100 g/mL relative to the GDA label (56). As noted above, the sodium warning label might not have the intended effect if it is displayed with other warning labels. Prior studies showed that multiple claims increase information costs [55] and confuse shoppers [21,56,57], thus reducing the effectiveness of any given claim [22]. However, we found that the high-in-calories and high-in-saturated fat labels worked even though the proportions of the products that displayed only these labels (22% and 11%, respectively) were smaller than those with only the sodium warning label (27%). It may be that, given the high prevalence of obesity in KSA, the sample participants were less concerned about sodium intake than about intake of calories and saturated fat. Indeed, in the baseline, among 651 participants who responded to read the NFP, only 3.4% chose sodium as the most important piece of information on the panel, which was a much smaller proportion compared with calories (50.1%), sugar (14%), and fats (8.1%). Lastly, given that our sample size was calculated for the primary outcome, the lack of power may be a potential explanation for the null effect of the WL on sodium.

Our findings come with the following limitations: First, our two cart criteria (i.e., a minimum expenditure and a minimum number of food categories in which to shop) probably reduced biases associated with hypothetical shopping, but the data for this study still represent hypothetical purchases, which may not represent actual purchases. Second, even if it provides an accurate reflection of purchases from a single shopping trip online, this study does not address the long-term effects of the FOP labels or the effects of the labels on offline grocery shopping patterns. Third, we modified the NS label, assuming that this change would increase its effectiveness for the targeted nutrient. Our results suggest this may be the case, but we did not conduct a formal test of this hypothesis because funding and other constraints did not allow for a four-arm trial. Fourth, although our primary outcome has been shown to be a valid measure of overall diet quality [38,39,40,41,42], we acknowledge concerns around the algorithm [37] (e.g., not considering mineral and vitamin content and high levels of “negative” components being possibly offset by the presence of “positive” components) and the possibility that another measure of diet quality might lead to different conclusions. Fifth, although we believe the WL messages were easily interpreted by participants, showing them an introductory video similar to what was shown to those in the NS arm might have generated a larger effect. Finally, our study results were based on a convenience sample in KSA and may not generalize to the general population. Despite these shortcomings, our findings are robust, providing useful information about the differential effects of two promising FOP labeling strategies. Although some of the baseline demographics were statistically different between those who completed the study and those who were randomized but not exposed to the interventions, we did not find evidence that those differences biased the study results, as our treatment effects were not affected by these covariates.

As managing risks for NCDs requires long-term behavior change, future studies should extend these results to test the labels over repeated shopping trips with actual purchases, both in online and in-store settings, and extend the analysis to measure clinical outcomes such as body mass index and blood pressure. In the real world, food purchase channels for home consumption are a mix of online and offline grocery shopping trips, where the effects of FOP labels may differ. Our results are applicable to similar online grocery shops where individuals make grocery purchases for future consumption, but the extent to which the labels improve diet quality may differ for other online shopping for more immediate consumption and when the range of products is more limited (e.g., online convenience store vs. grocery store). The results may also differ for in-store purchases, where visceral factors (sounds, smells, placement, and lighting) may attenuate the effects of the label. Future efforts should also formally test different variants of these labels to identify which is most beneficial for a given target population. For instance, given that Arabic is read from right to left, KSA shoppers may respond to the NS labels differently than those in other countries, as they read the sugar percentage before the grade for overall diet quality.

In conclusion, this study showed that both the modified NS labels and Chilean warning labels positively influenced food and beverage choices among KSA participants, but that there were differential effects across the two labels. The modified NS label with the sugar per serving tagline increased overall diet quality and decreased sugar per serving to a greater extent than the warning labels. By contrast, the warning labels were more effective at reducing calories and saturated fat per serving. Policymakers should consider these findings, along with their stated objectives, when determining which strategy may be best for their target population.

## Figures and Tables

**Figure 1 nutrients-15-02904-f001:**
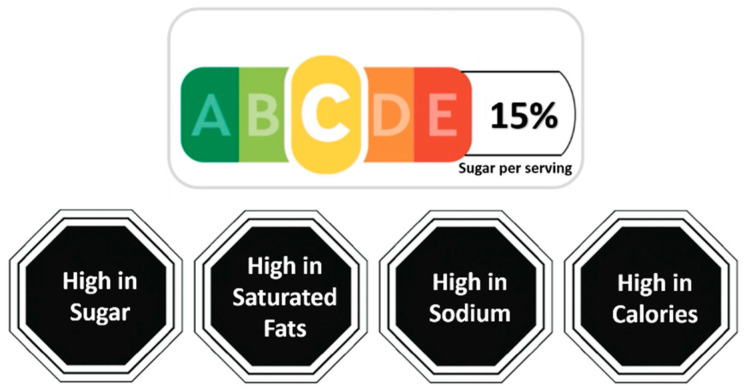
English-language versions of modified Nutri-Score labels (**Top**) and warning labels (**Bottom**).

**Figure 2 nutrients-15-02904-f002:**
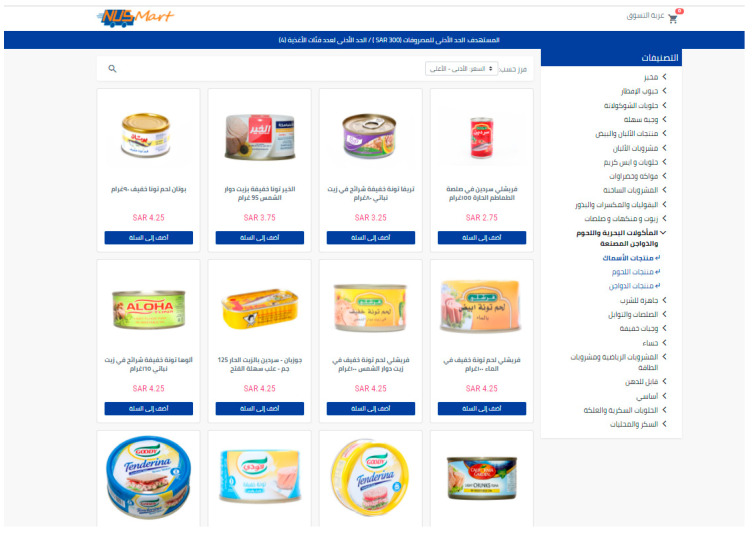
A screenshot of the experimental online grocery store for KSA (control arm).

**Figure 3 nutrients-15-02904-f003:**
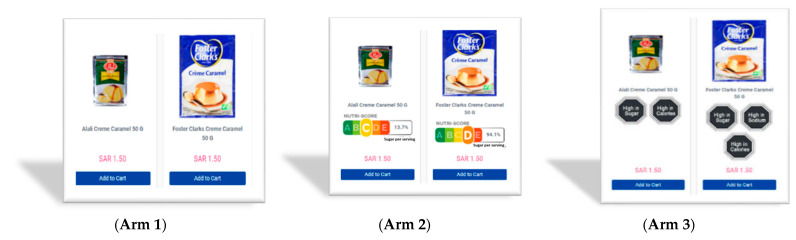
Example products from three versions of the online grocery store showing how the labels were presented across the three study arms; (**Arm 1**) (no-label control), (**Arm 2**) (NS), (**Arm 3**) (WL).

**Figure 4 nutrients-15-02904-f004:**
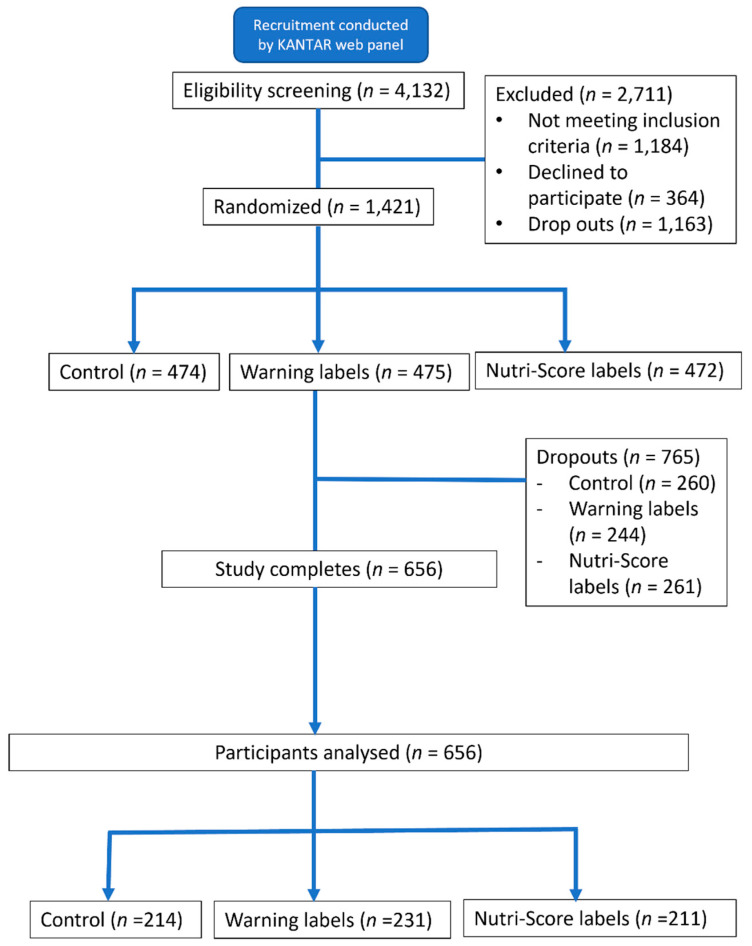
Participant flow diagram.

**Table 1 nutrients-15-02904-t001:** Summary statistics of the participants by arms and total (n = 656).

Demographic Variable	Control Arm	Warning Label Arm	NS Label Arm	Total
Age (years), mean ± SD	32.4 ± 6.5	31.3 ± 7.0	31.4 ± 7.0	31.7 ± 6.8
Female, (%)	56	41	50	49
Household Size, mean ± SD	4.9 ± 2.0	4.5 ± 2.0	4.6 ± 2.1	4.7 ± 2.0
High educational level (university degree and above), (%)	86	80	79	82
High household income (monthly income SAR 15,000 and above), (%)	46	45	44	45
Household with no underlying health condition, (%)	42	41	39	41
**Most Important Drivers of Shopping Behavior (%)**				
Taste	39	37	37	38
Price	16	19	16	17
Health	23	22	29	24
Variety	14	13	12	13
Convenience	8	9	6	8
Observations	214	231	211	656

Note: NS = Nutri-Score; SAR = Saudi riyals; SD = standard deviation.

**Table 2 nutrients-15-02904-t002:** Descriptive statistics of outcomes by arm (*N* = 656).

	Control	NS Arm	WL Arm
		Mean (SD)	
Weighted NS points per serving	32.3	34.8	32.8
	(4.1)	(4.9)	(4.8)
Energy (kcal) per serving	296.7	248.6	182.8
	(800.7)	(674.9)	(320.7)
Sugar (g) per serving	7.8	5.7	7.5
	(4.2)	(3.8)	(4.3)
Sodium (mg) per serving	787.3	673.1	862.2
	(2460.1)	(3123.6)	(5302.7)
Saturated fat (g) per serving	35.3	45.4	16.7
	(139.0)	(231.8)	(93.6)
Total energy (in 1000 kcal)	159.2	114.6	103.6
	(376.4)	(283.7)	(240.7)
Total sugar (kg)	4.5	2.9	4.0
	(4.7)	(2.7)	(3.7)
Total sodium (g)	726.1	306.4	573.9
	(2905.1)	(1053.1)	(3581.9)
Total saturated fat (kg)	24.7	18.5	9.0
	(107.1)	(87.8)	(42.2)
Observation	214	211	231

**Table 3 nutrients-15-02904-t003:** Effects of the NS and WL on weighted average NS point and per-serving energy and nutrients.

	Weighted NS Point	Energy (kcal) per Serving	Sugar (g) per Serving	Sodium (mg) per Serving	Saturated Fat (g) per Serving
NS	2.5 ***	−29.5	−2.1 ***	−120.1	6.1
	(0.4)	(68.6)	(0.4)	(246.5)	(15.9)
WL	0.6	−98.2 *	−0.5	28.8	−24.6 **
	(0.4)	(54.7)	(0.4)	(378.2)	(11.8)
NS vs. WL	1.9 ***	68.7	−1.6 ***	−148.9	30.7 *
	(0.5)	(52.0)	(0.4)	(389.5)	(17.4)

Notes: robust standard errors are in parentheses. * *p* < 0.1, ** *p* < 0.05, *** *p* < 0.01.

**Table 4 nutrients-15-02904-t004:** Effects of the NS and WL on total energy and nutrients of the shopping basket.

	Total Energy (in 1000 kcal)	Total Sugar (kg)	Total Sodium (g)	Total Saturated Fat (kg)
NS	−31.6	−1.4 ***	−373.3 *	−7.2
	(30.9)	(0.3)	(197.5)	(9.2)
WL	−40.8	−0.4	−110.7	−17.2 **
	(27.4)	(0.4)	(287.2)	(8.2)
NS vs. WL	9.1	−1.0 ***	−262.6	10.0
	(25.0)	(0.3)	(228.0)	(7.0)

Notes: robust standard errors are in parentheses. * *p* < 0.1, ** *p* < 0.05, *** *p* < 0.01.

**Table 5 nutrients-15-02904-t005:** Effects of the NS and WL on weighted average NS point and per-serving energy and nutrients (beverages only).

	Weighted NS Point	Energy (kcal) per Serving	Sugar (g) per Serving	Sodium (mg) per Serving	Saturated Fat (g) per Serving
NS	2.5 ***	−24.5 ***	−6.3 ***	−3.8	−0.0
	(0.6)	(6.2)	(2.00)	(5.1)	(0.1)
WL	1.6 ***	−17.9 ***	−5.0 ***	−2.7	−0.2 *
	(0.6)	(6.1)	(1.7)	(5.4)	(0.1)
NS vs. WL	0.9	−6.7	−1.3	−1.2	0.2
	(0.6)	(5.5)	(1.6)	(5.4)	(0.1)

Notes: robust standard errors are in parentheses. * *p* < 0.1, *** *p* < 0.01.

**Table 6 nutrients-15-02904-t006:** Effects of the NS and WL on total energy and nutrients of the shopping basket (beverages only).

	Total Energy (in 1000 kcal)	Total Sugar (kg)	Total Sodium (g)	Total Saturated Fat (kg)
NS	−1.3	−0.3	−0.6	−0.01
	(1.0)	(0.2)	(0.7)	(0.01)
WL	−1.2	−0.3	−0.5	−0.01
	(1.1)	(0.2)	(0.8)	(0.01)
NS vs. WL	−0.1	0.0	−0.1	−0.00
	(1.0)	(0.2)	(0.7)	(0.01)

Notes: robust standard errors are in parentheses.

## Data Availability

The datasets used and/or analyzed during the current study are not publicly available due to a lack of consent from all participants for making the data publicly available, but they are available from the corresponding author upon reasonable request.

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
