# Peer review of "A Randomized Controlled Study to Test Front-of-Pack (FOP) Nutrition Labels in the Kingdom of Saudi Arabia"

_nutrients, 2023, doi:10.3390/nu15132904_

Round 1
Reviewer 1 Report
The topic of the manuscript as well as the study design are very interesting and suitable to get a differentiate inside regarding FOP labels. Nevertheless, the introduction could be improved with a little more structure regarding the different kinds of labels, like 1) name existing approaches and assigned labels, 2) describe the mode of action for the different kind of labels (also the psychological influence on the consumer as well as pro & cons) and 3) comparison of the label. In addition, the last paragraphs of the introduction are describing the method ( measure of nutritional quality, label) and should be move there for a better understanding.
Only a small notice: Figure 1 describes left and right, but the labels are presented in above and below.
Author Response
The topic of the manuscript as well as the study design are very interesting and suitable to get a differentiate inside regarding FOP labels. Nevertheless, the introduction could be improved with a little more structure regarding the different kinds of labels, like 1) name existing approaches and assigned labels, 2) describe the mode of action for the different kind of labels (also the psychological influence on the consumer as well as pro & cons) and 3) comparison of the label. In addition, the last paragraphs of the introduction are describing the method (measure of nutritional quality, label) and should be move there for a better understanding.
We thank the reviewer for the positive comments. To address this concern, we rewrote paragraphs in the introduction to read as follows:
“FOP labels can generally be classified as reductive or interpretive. Reductive labels present a subset of relevant information without interpretation, such as presenting calories per serving and calories per day for a healthy diet. Interpretive labels use nutritional information to convey a message to consumers as to the underlying healthfulness of the product in the dimensions considered. Interpretive FOP labels can focus solely on identifying foods that are healthier (i.e., positive labels) such as in Singapore’s Healthier Choice Symbol (HCS) labels (19); on foods that are less healthy (i.e., negative labels), such as Chile’s warning labels (14); or both (i.e., graded labels), such as France’s Nutri-Score label (20), which grades all foods from A to E on overall nutritional quality. Which label is most effective at influencing overall diet quality for a given population is ultimately an empirical question. Further, each label is likely to have both intended and unintended consequences. For example, because a positive label offers a signal that a product is “healthier” (21), it could induce consumers to overconsume labeled products, thus resulting in an improvement in diet quality and, ironically, an increase in total calories (22). For this reason, countries have moved away from solely showing positive FOP labels in favor of graded approaches or warning labels. Graded labels, which condense the complicated information on the NFP into a single summary score, are likely to be more effective at improving overall diet quality (23, 24) than warning labels. This results because graded labels identify the healthiness of all foods. By contrast, warning labels only identify the worst offending foods to avoid, and often based only on a single nutrient, such as sugar or sodium. Not only do they not help consumers determine overall healthfulness of a given food, they also do not identify which of the non-targeted foods are healthier.”
Regarding the reviewer’s comment about the last two paragraphs, they briefly describe the hypotheses, outcomes, and experimental designs to test the hypotheses. It is common to present hypotheses with the study aims in the introduction and then to expand on the methods in the Methods section, as we do here. We propose to stick to this approach but will make these changes if the editor requests.
Only a small notice: Figure 1 describes left and right, but the labels are presented in above and below.
We noticed that the labels were presented vertically when the manuscript was re-formatted with the journal template. We have revised the caption of Figure 1, accordingly.
Reviewer 2 Report
The manuscript presents interesting data over conducted research.
All examination is designed in a correct way, even if the models dedicate to food labeling still cause many concerns (some described in the body of the manuscript).
Suggested division and description of the experiment is correct. However two issues should be solved.
1. Can authors point how much online shopping is influencing on consumer choices (vs traditional shopping)? Were consumers buying “more expected” (healthier) food, when they could suspect to be evaluated?
2. Authors evaluated consumer weight (according to declared?), how was it influencing on food choices vs “three study arm”?
Author Response
The manuscript presents interesting data over conducted research. All examination is designed in a correct way, even if the models dedicate to food labeling still cause many concerns (some described in the body of the manuscript). Suggested division and description of the experiment is correct. However, two issues should be solved.
- Can authors point how much online shopping is influencing on consumer choices (vs traditional shopping)? Were consumers buying “more expected” (healthier) food, when they could suspect to be evaluated?
The nature of our experiment does not allow us to directly test 1) consumer choices in online vs. offline stores and 2) the Hawthorne effect (i.e., people’s behaviors change when they become aware that they are being observed/evaluated). Regarding 1), however, prior studies showed consumers’ shopping behaviors are different between the two channels: they were shown to be less price sensitive, more brand loyal (i.e., stick to high market share brands without exploring other options), more size loyal (i.e., purchase in greater quantities) and to shop healthier when shopping online as opposed to offline [1-4]. We have added a limitation that this study does not address the effects of the labels on offline grocery shopping patterns.
Regarding 2), we do not think that the Hawthorne effect is a concern. First, we explained the general nature of the study tasks (i.e., grocery shopping) but did not reveal the aims and the testing hypotheses of the study (i.e., whether food labels promote healthier food choices) to participants until they completed the experiment. Second, even if the Hawthorne effect exists, this should exist across all arms, thus our between arm estimates are likely to be unbiased.
- Authors evaluated consumer weight (according to declared?), how was it influencing on food choices vs “three study arm”?
We estimated the same model in Section 2.4.2 but included BMI (weight (in kg)/height (in m) squared) for the primary outcome. We did not find a statistically significant effect of BMI on diet quality. We also included the interaction terms between BMI and dummies for the treatment arms but none of them were statistically significant.
References
- Chu J, Chintagunta P, Cebollada J. Research note—A comparison of within-household price sensitivity across online and offline channels. Marketing science. 2008 Mar;27(2):283-99.
- Chu J, Arce-Urriza M, Cebollada-Calvo JJ, Chintagunta PK. An empirical analysis of shopping behavior across online and offline channels for grocery products: the moderating effects of household and product characteristics. Journal of Interactive Marketing. 2010 Nov;24(4):251-68.
- Pozzi A. Shopping cost and brand exploration in online grocery. American Economic Journal: Microeconomics. 2012 Aug 1;4(3):96-120.
- Harris‐Lagoudakis K. Online shopping and the healthfulness of grocery purchases. American Journal of Agricultural Economics. 2022 May;104(3):1050-76.
Reviewer 3 Report
Please see the attachment.

Author Response
A brief summary:
This study aimed to test two promising Front-of-Pack (FOP) strategies by randomizing 656 KSA adults into one of the three versions of the store to complete a hypothetical grocery shop: No-label (control), NS (France’s Nutri-Score), and WL(Chilean warning label). Results of this study will help policymakers determine which strategy is likely to be most effective in KSA. The results showed that in terms of per-serving energy and nutrients, NS reduced sugar intake per serving, whereas WL reduced calories and saturated fat per serving relative to the control arm. None of the labels reduced sodium (mg) per serving. Both labels are effective at improving the nutritional quality of beverages purchased relative to the no-label control. These two labels were also effective at reducing calories and sugar.
Weakness of this study:
- Reducing sugar does not necessarily mean healthy. Artificial sweetener has been applied in more and more products such as soda, energy bar, even yogurt. Recent research studies have shown that artificial sweetener has related to insulin resistant, which has caused very bad health issues. So focusing on sugar intake and calories intake only makes this study weak in design.
Our sugar data captures both natural and added sugar content in the products, but it does not capture artificial sweeteners that often have zero calories such as aspartame. It is because we could not collect sugar substitute information from the Nutritional Facts Panel (NFP) printed on the back of the product packages. Although the association between the intake of sugar substitutes and chronic diseases such as diabetes and cancers has not been firmly established, we agree that the effects of food labels on sugar substitute changes should be explored in future studies.
- This study does not mention whether or not shoppers have options with regards to the items in store. For instance, if the only option for chicken can is the one with high sodium, even shoppers aware that high sodium is bad for health, they would buy anyhow because that is the only option they have.
We ensured that each food category contained a good number of options. As mentioned in the Methods section (page 3), there were, on average, 199 unique products per food category and 69 unique products per subcategory.
- This study does not consider shoppers above age 56. People’s health at this age stage is more at risk than young generations.
We acknowledged this limitation in the discussion section that our study results (i.e., changes in food choice patterns) were based on a convenience sample in KSA and may not generalize to the general population. However, given that participants, as primary grocery shoppers, were asked to choose groceries not only for themselves but their household members, the effects of their healthier food choices, if they were a real purchase, would be extended to their household members including children and the older adults above age 56 who were not included in the study.
- Websites in part 2.1 should be included in reference list.
We have included the websites in Section 2.1. in the reference list.